# Tracking the Pathways of West Nile Virus: Phylogenetic and Phylogeographic Analysis of a 2024 Isolate from Portugal

**DOI:** 10.3390/microorganisms13030585

**Published:** 2025-03-04

**Authors:** Diogo Maroco, Ricardo Parreira, Fábio Abade dos Santos, Ângela Lopes, Fernanda Simões, Leonor Orge, Sofia G. Seabra, Teresa Fagulha, Erica Brazio, Ana M. Henriques, Ana Duarte, Margarida D. Duarte, Sílvia C. Barros

**Affiliations:** 1Nacional Institute of Agrarian and Veterinarian Research, Quinta do Marquês, Av. da República, 2780-157 Oeiras, Portugal; diogompmaroco@gmail.com (D.M.); fabio.abade@iniav.pt (F.A.d.S.); angela.lopes@iniav.pt (Â.L.); fernanda.simoes@iniav.pt (F.S.); leonor.orge@iniav.pt (L.O.); teresa.fagulha@iniav.pt (T.F.); margarida.henriques@iniav.pt (A.M.H.); ana.duarte@iniav.pt (A.D.); margarida.duarte@iniav.pt (M.D.D.); 2Institute of Hygiene and Tropical Medicine, NOVA University, Global Health and Tropical Medicine (GHTM), Associate Laboratory in Translation and Innovation Towards, Global Health (LA-REAL), Rua da Junqueira 100, 1349-008 Lisbon, Portugal; ricardo@ihmt.unl.pt (R.P.); sgseabra@ihmt.unl.pt (S.G.S.); 3Medical Microbiology Unit, Institute of Hygiene and Tropical Medicine, NOVA University, Global Health and Tropical Medicine (GHTM), Associate Laboratory in Translation and Innovation Towards, Global Health (LA-REAL), Rua da Junqueira 100, 1349-008 Lisbon, Portugal; 4CECAV-Centro de Ciência Animal e Veterinária, Faculdade de Medicina Veterinária de Lisboa-Universidade Lusófona, Centro Universitário de Lisboa, 1749-024 Lisbon, Portugal; 5Global Public Health Unit, Institute of Hygiene and Tropical Medicine, NOVA University, Global Health and Tropical Medicine (GHTM), Associate Laboratory in Translation and Innovation Towards, Global Health (LA-REAL), Rua da Junqueira 100, 1349-008 Lisbon, Portugal; 6Wildlife Rehabilitation Centre of Lisbon (LxCRAS), Parque Florestal de Monsanto, Monte das Perdizes, 1500-068 Lisbon, Portugal; erica.brazio@cm-lisboa.pt; 7Associate Laboratory for Animal and Veterinary Sciences (AL4AnimalS), Avenida da Universidade de Lisboa, 1300-477 Lisbon, Portugal; 8Faculdade de Medicina Veterinária, Centre for Interdisciplinary Research in Animal Health (CIISA), Universidade de Lisboa, Avenida da Universidade Técnica, 1300-477 Lisbon, Portugal; 9Faculdade de Ciências e Tecnologia, Universidade Nova de Lisboa, Campus da Caparica, 2829-516 Almada, Portugal

**Keywords:** *Flavivirus*, West Nile virus, phylogenetic, phylogeographic, goshawk, birds, Portugal, whole-genome sequencing

## Abstract

Birds are natural hosts for numerous zoonotic viral pathogens, including West Nile virus, which is transmitted by mosquitoes. During migration, birds can act as vectors for the geographic spread of viruses. WNV is endemic in Portugal, causing annual outbreaks, particularly in horses. Here, we report the first detection of an avian WNV strain isolated from a wild bird (*Astur gentilis*) collected in Portugal in mid-September 2024. Phylogenetic and phylogeographic analyses were conducted to trace the virus’s origin and potential transmission routes, integrating the obtained full-length genomic sequence with a dataset of WNV strains from Africa and Europe (1951–2024). Phylogenetic analysis of 92 WNV sequences spanning lineages 1–5 positioned the 2024 isolate within lineage 1a. Results obtained using phylodynamics-based analysis showed that this isolate likely originated in Africa and reached Portugal via Spain’s Cádiz coast, confirming previously described WNV dispersal patterns between Africa and Europe. The data suggest a migratory route from West Africa to Europe, extending through countries such as Senegal, Mauritania, Morocco, Portugal, Spain, Italy, and France, indicating a reciprocal flow of the virus back into Africa. These transmission routes match the migratory paths of Afro-Palearctic bird species, emphasizing the role of migratory birds in the long-distance spread of WNV.

## 1. Introduction

West Nile virus (WNV) is an arthropod-borne enveloped virus classified in the *Orthoflavivirus* genus of the family *Flaviviridae*. Its genome comprises approximately 11 kb of positive-sense single-stranded RNA (ssRNA+), containing a single open reading frame (ORF). This ORF encodes a polyprotein precursor that is synthesized at the rough endoplasmic reticulum and is subsequently cleaved into three structural proteins: C (capsid), prM (membrane protein precursor), and E (envelope), and seven non-structural proteins: NS1, NS2A, NS2B, NS3, NS4A, NS4B, and NS5 [1].

WNV is primarily transmitted through a cycle involving birds and *Culex* sp. mosquitoes, while in some cases, humans and other animals are accidental or “dead-end hosts”. Unlike birds, humans and most mammals are considered “dead-end hosts” because the level of viremia is usually not high enough for mosquitoes to become infected [1,2,3].

To date, at least nine different WNV lineages have been identified, with the majority of reported human infections being associated with lineages 1a (L1a) and 2 (L2) [4,5,6]. Most of these infections are asymptomatic, with only a small percentage resulting in West Nile fever—a flu-like illness—and, in very rare cases, severe neurological complications that can lead to fatal encephalitis, especially in elderly and immunocompromised individuals [7,8].

Since its first identification in the West Nile district of Uganda in 1937 [9], WNV has spread widely across the globe. In Europe, the largest WNV epidemic was recorded in 2018 [10], with reported cases exceeding the total number of infections documented since 2010 [11]. Later on, in 2020, countries such as the Netherlands and Spain experienced significant outbreaks [12], with a considerable number of human infections and deaths reported, even in previously unaffected regions. The most recent data from the European Centre for Disease Control (ECDC; https://www.ecdc.europa.eu/en (accessed on 17 December 2024) indicate that, by December 2024, 19 European countries reported a total of 1436 locally acquired human cases of WNV infection. These cases are significantly higher than the monthly average over the past 10 years and exceed the 751 human cases reported in 2023 [11].

In animals, a total of 494 outbreaks of WNV in equids and 447 outbreaks in birds were reported in Europe in 2024, with the majority of avian cases occurring in Italy [13].

So far, only four serologically confirmed human cases have been reported in Portugal. The first two were reported in 2004, when two Irish birdwatchers became infected during a trip to the southern province of Algarve [14]. In conjunction with the establishment of an emergency surveillance plan by the Portuguese health authorities, the virus from lineage 1a was detected in pools of *Culex pipiens* and *Cx. univittatus* mosquitoes collected from the same area. The viruses detected were isolated in Vero cells and sequenced [15,16]. Later, in 2010 and in 2015, two additional distinct human cases of infection were diagnosed, one from the district of Setúbal and the other in the Algarve region, respectively [17].

Following these human cases, several horses and birds have shown neutralizing antibodies to WNV as part of a surveillance effort [18,19]. The serological data from the past decade indicate that WNV has been circulating annually in the equine population of Portugal since at least 2015. During this period (2015–2024), there have been a total of 59 confirmed cases (evidenced by the presence of positive IgM antibodies), suggesting active circulation of the virus. In addition, 2024 was recorded as the worst year for WNV outbreaks in horses (*n* = 32), which was associated with a high mortality rate among these animals (INIAV, unpublished data). Despite numerous attempts to molecularly detect WNV from different samples originating from horses and birds, only serologically positive animals had been identified in the country. This limitation hindered the identification of the genomic lineage of the viruses involved in the annual outbreaks and the study of the evolutionary relationships and epidemiology of the infection.

Here we report the first detection of an avian WNV isolated from a wild bird (*Astur gentilis*), collected in mainland Portugal in mid-September 2024. The entire genome of this isolate from lineage 1a was fully sequenced, and phylogenetic and phylogeographic analyses were conducted to elucidate its origin and potential transmission routes. This was achieved by integrating its genomic sequence in a dataset that includes previously sequenced WNV strains from Africa and Europe, spanning the years 1951 to 2024, with a particular focus on lineage 1a isolates.

Through this work, we aimed to gain a deeper understanding of WNV’s circulation between these two continents, provide an updated dataset that will facilitate further studies on the dispersal/spread dynamics of lineage 1a strains, and highlight the importance of proactive and effective surveillance strategies for WNV. Such strategies would improve early warning systems, risk reduction efforts, and increase our preparedness for this emerging zoonotic threat in an era of increasing climate change and globalization.

## 2. Materials and Methods

### 2.1. Case Description

A European Goshawk (*Astur gentilis*), a resident wild bird, was found alive but in an under ideal body condition (score 1 in a 5 scale) in a port area of the municipality of Setúbal (GPS coordinates 38.489, −8.789). The bird was a juvenile female specimen weighing only 705 g (adult females can weigh up to 1500 g). Clinical signs observed included motor incoordination, closed eyes, lethargy, emaciation, and dehydration. It was collected and transported by official entities to the Wildlife Rehabilitation Centre of Lisbon (LxCRAS) for rescue and treatment, where it arrived on September 6 and died shortly thereafter, on 7 September. The bird’s carcass was sent to the Portuguese National Institute of Agricultural and Veterinary Research (INIAV I.P.) as part of an active avian influenza (AI) surveillance program implemented in the country.

### 2.2. Histopathological Description

For histopathological examination, a fragment of brain was fixed in 10% buffered formalin and embedded in paraffin using standard procedures. Then, 5 μm-thick sections were stained with hematoxylin and eosin (H&E) and examined using light microscopy (Olympus BX60 (Olympus, Tokyo, Japan) and Image Focus Alpha^®^ camera (Euromex, Duiven, The Netherlands).

### 2.3. Nucleic-Acid Extraction and Viral RNA Detection

Brain, spleen, and liver samples from the deceased goshawk were individually homogenized at a concentration of 20% (*w/v*) with phosphate-buffered saline by mechanical maceration with 0.5 mm zirconium beads (Sigma-Aldrich, St. Louis, MO, USA) using four cycles of 15 s at 3000 rpm with a 10 s interval between each cycle (Bertin Technologies, Paris, France). The homogenates were then clarified by centrifugation at 3000 *g* for 5 min at 4 °C.

Total nucleic acid extraction was performed using 200 μL of the clarified supernatant, employing the IndiMag^®^ Pathogen Kit (Indical, Leipzig, Germany) with a KingFisher Flex extractor (ThermoFisher Scientific, Waltham, MA, USA), following the manufacturer’s instructions. The extracted nucleic acids were stored at −20 °C until use. The extraction process was validated using an 18S rDNA qPCR [20] and an RT-qPCR for the detection of spiked synthetic RNA (VLP-RNA EXTRACTION, Meridian Life Science, Memphis, TN, USA), which was added to the sample (4 µL/sample) prior to extraction. Simultaneously, as a negative control for the extraction process, a negative sample was subjected to the same treatment as the experimental samples, without any biological material.

The presence of the avian influenza virus genome was screened using a previously described real-time quantitative reverse transcription PCR (RT-qPCR) method [21].

Detection of the WNV genome was performed using RT-qPCR targeting the NS2A gene (Primer Fw: CCTTTTCAGYTGGGCCTTCTG; Primer Rv: CAGTGTAVGTVATRCCCCCAA; Probe FAM—AGCCAAGATCAGCATGCCAGC) as described by [22], utilizing the One-step NZYSpeedy RT-qPCR Probe kit (NZYtech, Lisboa, Portugal) in a 20 μL reaction volume. A positive control reaction was prepared in parallel, using RNA from the Egypt 101 strain (accession number EU081844). Similarly, a negative control reaction was prepared following the same protocol, replacing the total volume of extracted RNA with RNAse-free water. Amplifications were carried out in a Bio-Rad CFX96™ Thermal Cycler (Bio-Rad Laboratories Srl, Redmond, WA, USA). Cycle threshold (Ct) values over 38 were considered negative.

### 2.4. Virus Isolation

The goshawk brain sample with the lowest Ct value was selected for virus isolation. This sample was homogenized at a concentration of 20% (*w/v*) in PBS containing penicillin, streptomycin, and amphotericin B (antibiotic-antimycotic), according to the manufacturer’s instructions (Gibco, Waltham, MA, USA). After centrifugation (3000 *g* for 10 min at 4 °C), the supernatant was filtered through a 0.45 μM pore size filter (Millipore Express, Darmstadt, Germany) and used to inoculate sub-confluent (70%) Vero cells (ATCC CCL-81), which were maintained in Dulbecco’s Modified Eagle Medium supplemented with 5% fetal bovine serum (FBS), penicillin, streptomycin, and amphotericin B (antibiotic-antimycotic used at a ratio of 1:100; ThermoFisher Scientific, Waltham, MA, USA), along with 50 µg/mL gentamicin (ThermoFisher Scientific, Waltham, MA, USA). The cells were incubated at 37 °C in a humidified atmosphere with 5% CO_2_ and monitored daily for cytopathic effects (CPE) using phase-contrast microscopy.

### 2.5. Sample Preparation for Next-Generation Sequencing

To prepare samples for MinION whole-genome sequencing, the supernatant of WNV-infected Vero cells (third passage) was clarified by centrifugation at 3220 *g* for 10 min, then filtered through a 0.45 µM membrane filter (Sigma-Aldrich, St. Louis, MO, USA) and treated with 10 U of TURBO DNase (Thermo Fisher Scientific, Waltham, MA, USA) for 1 h at 37 °C. Virus concentration was achieved by ultracentrifugation at 91.500 *g* for 2 h. Viral RNA was extracted using the IndiMag^®^ Pathogen Kit (Indical, Leipzig, Germany) in a Kingfisher Flex extractor (Thermo Fisher Scientific, Waltham, MA, USA) according to the manufacturer’s protocol.

Complementary DNA (cDNA) synthesis was performed in duplicate by adapting the SISPA protocol described by [23], using the SuperScript IV First Strand Synthesis System kit (Thermo Fisher Scientific, Waltham, MA, USA). The extracted RNA was first annealed by preparing the following mixture: 5 μL of primer FR26RV-N (GCCGGAGCTCTGCAGATATCNNNNNN) [24] at 10 mM, 1 μL of dNTP mix (10 mM), 5 μL of RNA, and 2 μL of nuclease-free water. The reaction mixture was then incubated for 5 min at 65 °C before being cooled briefly on ice.

First-strand synthesis was performed by adding 4 μL of 5X RT Buffer, 1 μL of DTT (100 mM), 1 μL of SuperScript IV RT (200 U), and 1 μL of ribonuclease inhibitor (40 U). The final mixture was incubated at 23 °C for 10 min, followed by 50 °C for 45 min and 80 °C for 10 min, in a T100 Thermal Cycler (Bio-Rad Laboratories, Hercules, CA, USA). To eliminate any residual RNA molecules, 1 μL of *E. coli* RNase H (2 U) was added to the mixture, which was then incubated at 37 °C for 20 min.

To complete the formation of double-stranded cDNA molecules, 1 μL (5 U) of DNA Polymerase I Large (Klenow) Fragment (New England Biolabs Inc., Ipswich, MA, USA), 2.5 μL of 10X NEBuffer 2 (New England Biolabs Inc., Ipswich, MA, USA), and 0.5 μL of dNTP mix (10 mM) were added to 20 μL of cDNA. The final mixture was incubated at 25 °C for 15 min, followed by enzyme inactivation through the addition of 1 μL of EDTA (10 mM) and incubation at 75 °C for 20 min.

The dsDNA was amplified through PCR using the Speedy NZYTaq 2× Colourless Master Mix PCR kit (NZYtech, Lisboa, Portugal). The 50 μL reaction mixture was prepared as follows: 5 μL of dsDNA, 25 μL of 2X Speedy NZYTaq 2× Colourless Master Mix, 10 mM of primer FR20RV (GCCGGAGCTCTGCAGATATC) [24], and 15 μL of nuclease-free water. The thermal amplification profile included an initial denaturation step at 95 °C for 2 min, followed by 30 amplification cycles including 30 s for DNA denaturation at 95 °C, 20 s of primer hybridization at 55 °C, and a 2 min extension at 72 °C. A final extension step was also included, with a 10 min incubation at 72 °C. All PCR products were visualized by agarose gel electrophoresis, quantified using a NanoDrop™ 2000 Spectrophotometer (Thermo Fisher Scientific, Waltham, MA, USA), and stored at −20 °C.

### 2.6. Next-Generation Sequencing and Data Analysis

For MinION sequencing, libraries were prepared using the Rapid Barcoding Sequencing Kit (SQK-RBK114.24), following the manufacturer’s protocol (Oxford Nanopore Technologies, Oxford, UK). Sequencing was performed on a MinION Mk1B device with FLO-MIN114 flow cells for 3 h, employing a high-accuracy base calling model in the MinKNOW v24.11.10 software, with a minimum quality score (Q-score) threshold of 9.

Simultaneously, viral DNA libraries for Illumina sequencing were prepared from the same SISPA products, using the Illumina DNA Prep PCR free (Illumina) in conjunction with the Illumina DNA/RNA UD Indexes Set D (Illumina). All four main steps of the protocol—tagmentation, post-tagmentation clean-up, PCR amplification, and post-amplification clean-up—were performed according to the manufacturer’s instructions. The DNA concentration of each library was measured using the Qubit 4 fluorometer (Thermo Fisher Scientific, Waltham, MA, USA) to prepare equimolar pools for sequencing. Pooled sequencing libraries were sequenced bidirectionally using the MiSeq platform with the 2 × 150 bp v2 Reagent Kit.

The raw nucleotide reads were analyzed and assembled using the Genome Detective Virus Tool (https://www.genomedetective.com (accessed on 4 November 2024) through de novo and reference-based workflows.

### 2.7. Datasets, Phylogenetic and Phylogeographic Analyses

Two types of full-length WNV sequences datasets were used in this study. Both datasets included the WNV full-length sequence obtained through this work, as well as various WNV genomic sequences sourced from the NCBI Virus database (https://www.ncbi.nlm.nih.gov/labs/virus/vssi/#/ (accessed on 5 November 2024). Reference sequences that consisted of near-full-length genomes or full-length sequences with homopolymeric stretches of “N” were excluded from both datasets. One dataset primarily comprised L1a and L2 viral genomes, which are the most commonly found in Europe. However, it also included lineage 1b (Kunjin virus) and lineage 3–5 viral sequences, totaling 92 sequences (dataset 1). This dataset was exclusively used for phylogenetic inference using the maximum likelihood (ML) optimization criterium (as detailed below). The second dataset consisted of 82 lineage 1a ORF sequences (dataset 2) that were used for phylodynamic analysis. In both cases, the nucleotide sequences that made up each dataset were aligned using the MAFFT v7 software [25] and then manually edited to ensure that they were all aligned codon by codon. Each dataset was assessed for its phylogenetic signal by likelihood mapping using the TREE-PUZZLE v5.3 program [26].

The phylogenetic reconstruction conducted under the ML optimization criterium used the evolutionary model recommended by the IQ-TREE v2.4.0 software [27], assuming Akaike’s information criterium. The topological stability of the resulting tree was evaluated through conventional bootstrapping, with 1000 replicates of the original input data.

The temporal signal of the dataset used for phylogeographic reconstruction was tested using the TempEst v1.5.3 program [28], analyzing the temporal spectrum of divergence over time (from root to tip), and conducting a regression analysis to determine the R^2^ parameter.

Phylogeographic sequence dispersal in a continuous space was performed using the BEAST v1.10 software [29], assuming an uncorrelated lognormal relaxed molecular clock, rejecting a strict clock model based on the results of the molecular clock test incorporated into the MEGA X v11.0.13 software [30].

Phylodynamic analysis was carried out assuming both a parametric (constant) and the non-parametric Skyline coalescent priors and that sequence dispersal over the geographical space followed the Cauchy distribution model. Five independent Markov chain Monte Carlo (MCMC) chains were run in parallel for 300 million generations, with a sampling frequency of 300,000 and a burn-in of 10%. From these, results from three runs demonstrating optimal convergence for each model were combined using the LogCombiner program [29]. This process ensured chain convergence and adequate mixing, with the effective sample size (ESS) for all parameters exceeding 200, as verified using the Tracer v1.7.2 program [31]. A single maximum clade credibility tree, summarizing the information from the entire sample of phylogenetic trees, was generated using the TreeAnnotator program [29]. The resulting tree was visualized and edited with the FigTree v1.3.1 program. Finally, the phylogeographic sequence dispersal reconstruction was visualized using the SpreaD3 program [32].

### 2.8. Shannon Entropy and Selective Force Analyses

The ratio of non-synonymous to synonyms substitutions (dN/dS) was estimated using two tools: The synonymous non-synonymous analysis program (SNAP v.2.1.1) (https://www.hiv.lanl.gov/content/sequence/SNAP/SNAP.html (accessed on 9 January 2025)), which is based on the method described by Nei and Gojobori [33], and the single-likelihood ancestor counting (SLAC) (https://www.datamonkey.org/slac (accessed on 9 January 2025)), which applies a modified version of the Suzuki–Gojobori counting method [34].

To assess the variability and frequency of amino acids at each position in the polyprotein sequence, the Shannon entropy function was employed using the Shannon Entropy-Two tool (https://www.hiv.lanl.gov/content/sequence/ENTROPY/entropy.html (accessed on 9 January 2025)). This analysis was performed on a multiple sequence alignment comprising the 82 L1a sequences used in the phylogenetic reconstruction. In this approach, the 30 sequences most similar to the 2024 Portuguese isolate were defined as the “Query”, while the remaining 52 sequences from the dataset were assigned as the “Background” group. The Immune Epitope Database & Tools (IEDB) platform (https://www.iedb.org/ (accessed on 14 January 2025)) was also used to further investigate the correlation between observed positive selection sites acting on specific amino acid positions and previously described WNV epitopes. Additionally, the platform’s Bepipred Linear Epitope Prediction v2.0 tool (http://tools.iedb.org/bcell/ (accessed on 14 January 2025)) was used to predict the antigenic potential of these sites.

## 3. Results

### 3.1. Genome Detection and Virus Isolation

Screening for avian influenza virus by RT-qPCR [21] was negative in all tissues tested. The WNV genome was detected in RNA samples extracted from the brain, spleen, and liver of the goshawk by RT-qPCR [22], with Ct values of 16.59, 22.84, and 26.21, respectively.

The virus (WNV/18665/PT2024) was then successfully isolated in Vero cells from the brain tissue (sample with the lowest Ct value), using standard viral isolation procedures. The cytopathic effects (CPE), i.e., cell rounding and ballooning followed by detachment, were identified 3 days after the 2nd passage.

### 3.2. Histopathological Examination

Despite the presence of artefacts caused by freezing, intense congestion was observed in the meninges and neuroparenchyma, as well as a multifocal perivascular cuffing of mononucleated inflammatory cells in the neuroparenchyma of a cerebrum fragment, corresponding to a non-suppurative encephalitis (Figure 1).

### 3.3. Genome Sequencing

The successful amplification of viral cDNA using the SISPA protocol was evaluated through agarose gel electrophoresis. Duplicate samples were purified and quantified, yielding concentrations of 148.4 ng/μL and 155.8 ng/μL respectively, enabling the sequencing process to advance. This resulted in cumulative data from both Illumina and MinION sequencing encompassing 503,905 reads with an average coverage depth of 2208.

The results obtained from both MinION and Illumina sequencing methods were consistent, yielding nucleotide sequences that were 100% identical. This extensive data allowed us to confidently assemble the complete genome sequence, which was submitted to GenBank/ENA/DDBJ under the accession number PQ660506.

### 3.4. Phylogenetic and Spatiotemporal Analyses

The phylogenetic/phylogeographic analyses reported here were carried out using datasets characterized by high phylogenetic signal assessed by analyzing the topological resolution of the quartets of nucleotide sequences, randomly sampled from these datasets using likelihood mapping. In both cases, the obtained results showed a great majority of these quartets (97.360% for dataset 1 and 95.560% for dataset 2), supporting the reliability of these dataset for phylogenetic (Appendix A) and phylogeographic inference (Appendix A/Constant demographic prior and Figure 2/Bayesian Skyline coalescent prior).

The initial phylogenetic reconstruction, based on the dataset of 92 sequences spanning lineages 1–5, placed the 2024 WNV isolate within lineage 1a. This allowed a more focused analysis of the phylogenetic relationships exclusively among members of this lineage (Appendix A). A new phylogenetic reconstruction was then performed using the second dataset containing only 82 L1a sequences (Figure 2). Root-to-tip regression analysis indicated a strong correlation between genetic distance and sampling dates, demonstrating a robust molecular clock signal (R^2^ = 0.843) and an estimated evolutionary rate of 4.268 × 10^−4^ substitutions/site/year. This value is almost identical to the mean rate value of the BEAST analysis (4.563 × 10^−4^), i.e., the overall mean substitution rate across the entire tree, while the ucld.rate (i.e., the mean of the uncorrelated lognormal distribution used in the relaxed clock model drawn from a lognormal distribution) was slightly higher (8.538 × 10^−4^). In all cases, these results are consistent with previously reported rates for RNA viruses, and in particular for WNV [35,36,37,38].

The analysis of the lineage 1a WNV spatiotemporal dispersal between Africa and Western Europe gave essentially congruent results regardless of the coalescent prior used. As recently reported [39], the dispersion of most of the WNV L1a strains that have been circulating in Europe, and which have been found in Portugal, Spain, and Italy, seem to share common ancestry with others of West African origin, involving countries such as Senegal (the majority) and Morocco. However, also associated with the countries mentioned above, a smaller number of viral strains seem to have an ancestral origin centered in Egypt. Altogether, our analysis confirms the above-mentioned study, indicating a North African origin for the common ancestor of all the WNV strains under analysis with high posterior probability (pp) dating from the early 20th century (pp = 1). In addition, the phylogeographic dispersal analysis suggested that the 2024 WNV isolate from Portugal shared common ancestry with an ancestor originating from the African continent, with Senegal identified as its most likely country of origin (pp = 1). The findings also suggested that a common ancestor of the 2024 WNV isolate first arrived in the Iberian Peninsula around 2018 (95% HPD: 2014–2021), entering via Cádiz, Spain. This common ancestor appears to have diverged around 2020 (95% HPD: 2016–2022), spreading in two distinct directions: one towards mainland Portugal, resulting in the 2024 Portuguese isolate, and the other towards central Spain, where the 2022 Spanish strains were subsequently established. This pattern of dissemination provides a plausible explanation for the observed genetic relatedness between the Portuguese WNV isolate and the 2022 Spanish strains.

### 3.5. Shannon Entropy and Selective Force

The Shannon entropy results, based on the analysis of aligned ORF amino acid sequences and summarized in Table 1 and Figure 3, indicated a relatively overall low entropy concerning the individual amino acids present in the polyprotein sequences. This finding suggested that amino acid sequences of the different viral proteins have remained largely unchanged over the years, despite the global spread of this virus. However, the “Query” group, corresponding to the 30 sequences most similar to the 2024 WNV Portuguese isolate, showed higher entropy than the remaining 52 sequences in the dataset at 10 different positions while, on the other hand, relatively lower entropy values were associated with eight different positions. Most of these differences in entropy between the background and the query were observed in regions of the amino acid sequence associated with the non-structural proteins, while two other affected either the capsid or the envelope structural proteins.

Furthermore, the corresponding alignments of viral genomic nucleotide sequences were also analyzed with the SNAP and SLAC methods, which infer nonsynonymous (dN) and synonymous (dS) substitution rates on a per-site basis using a combination of maximum likelihood (ML) and counting approaches. The SNAP tool estimated a dS/dN ratio of 70.953, highlighting a strong predominance of synonymous over nonsynonymous substitutions within this group of sequences. This suggested an evolutionary process whereby polyprotein sequences tend to remain nearly identical, i.e., under a strong overall negative selection.

The data obtained using the SLAC method also showed evidence of widespread negative selection at 510 sites at a *p*-value threshold of 0.1. However, a *p*-value of 0.01 indicated pervasive positive selection at a site corresponding to the 1754th codon, which is part of the NS3 coding region (position 249 in the NS3 protein). At this site, sequences from the dataset predominantly have either a proline (P) or a threonine (T), with only two sequences (GQ851606 and AY701412) showing an alanine (A). Using the IEDB platform, a characterized WNV epitope associated with this amino acid position was identified (IEDB ID: 167889). This linear epitope (EALRGLPIRYQTSAVPR) was evaluated for immune reactivity and has showed qualitative T cell binding in two different assays [40,41].

By analyzing an amino acid sequence from the NS3 coding region (PGAGKTRRILPQIIKEAINRRLRTAVLAPTRVVAAEMAEALRGLPIRYQTSAVTREHNGNEIVDVMCHATLTHRLMSPHRVPNYNLFVMDEAHFTDPASIAARGY), the Bepipred Linear Epitope Prediction v2.0 tool also identified this segment as a likely antigenic site with a high potential for B cell binding (Figure 4).

Notably, the substitution of P to T in this position, observed in the 2024 Portuguese WNV isolate, resulted in a decrease in the residue prediction score, falling from 0.576 to 0.554. Although this reduction is relatively modest, these results further highlight the antigenic potential of this region for both T and B cells.

Furthermore, this specific amino acid substitution and subsequent changes may affect mechanisms of immune evasion.

## 4. Discussion

For several years, the Portuguese National Laboratory for Veterinary and Agrarian Research has been actively monitoring WNV in bird samples, most of which are submitted to the laboratory as part of the national avian influenza surveillance program managed by the Portuguese veterinary authorities. Positive WNV serology has been previously registered in birds [18]. However, despite testing a high number of bird samples for WNV genome detection over the years, a positive WNV RT-qPCR result was only obtained in 2024 from a European goshawk initially suspected to be infected with avian influenza virus.

The lesions observed, including meningeal and neuro parenchymal congestion, multifocal perivascular cuffing of mononuclear inflammatory cells, and non-suppurative encephalitis, provide a clear pathophysiological basis for the progressive weight loss and subsequent death of the animal. The neuroinflammation caused by WNV likely disrupted normal neurological function, leading to impaired motor coordination, reduced ability to hunt or feed, and systemic metabolic dysregulation. Additionally, the chronic inflammatory response in the central nervous system would have contributed to a state of hypercatabolism, further exacerbating the falcon’s weight loss. These combined factors, alongside the potential for secondary infections due to immunosuppression, ultimately overwhelmed the bird’s physiological resilience, resulting in death.

This study enabled the isolation and complete genome sequencing of the 2024 WNV strain from Portugal using two next-generation sequencing (NGS) methods. The resulting data allowed the full genetic characterization of the isolate (WNV/18665/PT2024) and, through phylogenetic and phylogeographic analyses, provided insights into its origin, genetic relationships, and the dispersal dynamics of L1a strains between the African and European continents. Overall, the obtained results agree with a recently published analysis of WNS L1a dispersal between Africa and Europe [39].

The widespread distribution of WNV and the ecology of its enzootic transmission cycle—closely linked to avian host migration patterns, arthropod vector dynamics, and climate change—highlight the importance of phylogenetic and phylogeographic studies. These analyses offer valuable insights into the distribution and evolutionary history of the pathogen, improving our understanding and ability to predict and refine its epidemiology.

The identification of WNV/18665/PT2024 infecting a goshawk, a resident wild bird, rescued at the district of Setúbal is particularly significant due to the ecological characteristics of the region, which include its extensive coastline, proximity to the sea and freshwater bodies, and its characteristic Mediterranean climate. These ecological features make the region particularly vulnerable to the maintenance and reproduction of mosquitoes, while also facilitating interactions between mosquitoes and both marine and migratory birds. In fact, from 2004 onwards, neutralizing antibodies against WNV have been detected in horses and birds originating from this high-risk area for WNV infection [18,19]. Together with molecular evidence, our findings suggest ongoing WNV circulation in Portugal, particularly concentrated in the coastal regions of southern and south-western Portugal, as previously documented [42].

Southern European countries, such as Portugal, are particularly vulnerable to the emergence of arboviruses due to their Mediterranean climate and the ecological richness of their wetlands. These wetlands serve as important stopovers for Palearctic migratory birds traveling between Eurasia and Africa [43,44]. Such regions provide ideal conditions for the overwintering and amplification of mosquito populations, thereby increasing the likelihood of WNV transmission to humans and other large mammals, such as horses, which are particularly susceptible to WNV.

The phylogenetic and phylogeographic data obtained from this study have not only indicated the African origin of the 2024 Portuguese WNV isolate but also revealed its ancestral route to Spain via the coastal region of Cádiz, which seems to have occurred in the last decade. It also confirmed existing descriptions of the dynamics of WNV spread between these two continents [39,45]. The evidence gathered illustrates a pattern of WNV circulation from West Africa to Europe and, more importantly, from Europe back to the African continent. This suggests a pathway extending from Senegal through Mauritania, Morocco, and many western and southern European countries, including Portugal, Spain, Italy, and France, which has been supported by previous studies [39,46,47,48]. These routes into and out of Europe also appear to be consistent with the documented migratory flyways of several Afro-Palaearctic bird species [49,50,51,52], highlighting the important role of migratory birds as carriers in the spread of WNV over large geographical distances.

Shannon entropy and selective pressure analyses provided significant insight into the evolutionary dynamics of the amino acid sequences corresponding to the L1a dataset (dataset 2). Shannon entropy results indicated general low entropy associated with most codons. However, 10 amino acid positions in the “Query” dataset that included the WNV/18665/PT2024 strain displayed higher entropy than the used background, while for 8 other positions, the entropy was lower than expected, indicating high conservation. Moreover, both analytical methods used for assessment of selective pressure on WNV evolution indicated that the great majority of substitutions in these sequences were synonymous, and did not affect the final polyprotein sequence. The observed selective pressure was, therefore, predominantly negative, suggesting that the evolution of the WNV is biased towards conservation. This stability may be related to the replication strategy of the virus, which depends on various post-transcriptional and post-translational modifications that are highly sensitive to amino acid changes that could affect viral protein activity and stability [53,54,55]. However, our results identified one substitution under strong positive selection pressure within the NS3 coding region. Using the IEDB database and tools, this site was associated with a known T-cell epitope [40,41] and showed a significant likelihood of being a B-cell epitope. These findings may suggest that the observed substitution may contribute to immune evasion. However, further studies are needed to evaluate the impact of this substitution on the immune recognition of this antigenic site.

Despite these advances, significant knowledge gaps remain due to the scarcity of molecular data, which hampers our understanding of the epidemiology and the prediction of WNV transmission pathways. For example, the four-year interval between the arrival of the ancestor in the Iberian Peninsula and its detection in Portugal in 2024 represents a missed opportunity to clarify dissemination patterns.

Furthermore, given that this study is based on a single isolate, this constitutes a limitation, as this sample may not accurately represent the diversity of strains present in circulation within Portugal. This underscores the importance and urgency of establishing robust and sustained longitudinal surveillance systems.

## Figures and Tables

**Figure 1 microorganisms-13-00585-f001:**
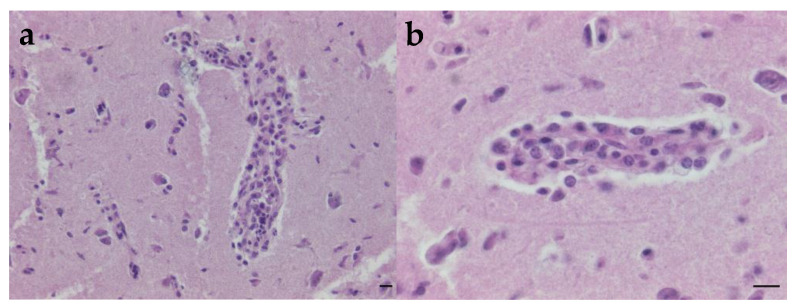
Mononuclear cell perivascular cuffing in grey matter in the cerebrum in two different fields of view ((**a**,**b**), H&E, bar = 10 µm).

**Figure 2 microorganisms-13-00585-f002:**
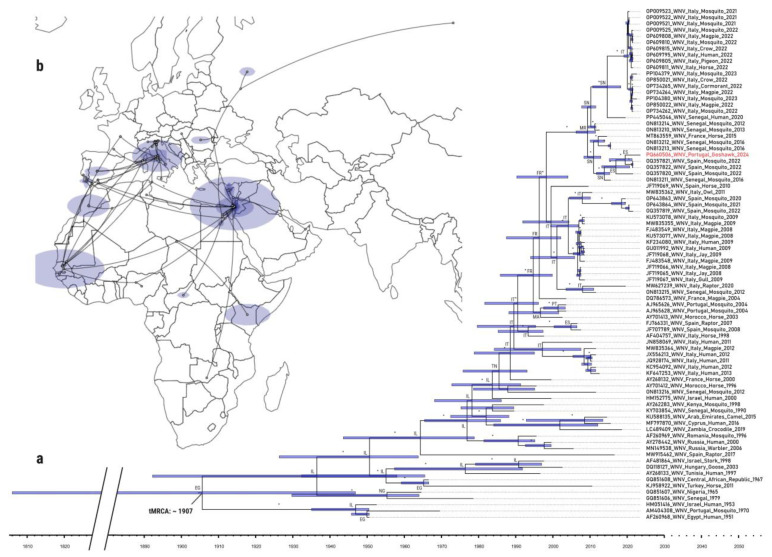
(**a**) Time-scaled Bayesian phylogenetic tree of West Nile virus lineage 1a strains. Posterior probabilities at specific branch nodes that are equal or greater than 0.85 are indicated by an asterisk (*). The 95% highest posterior density (HPDs) of the median node ages are represented by blue bars. The time of the most recent common ancestor (tMRCA) was estimated to be around 1907. Viral sequences are identified by their accession number, country of origin, host, and year of collection. The WNV/18665/PT2024 genomic sequence is indicated in red. The estimated locations of specific branch nodes are shown using their respective ISO 3166-1 alpha-2 country codes. (**b**) Phylogeographic reconstruction and diffusion pattern of the 82 L1a genomes and their ancestors. The blue polygon areas correspond to the probability density associated with those locations.

**Figure 3 microorganisms-13-00585-f003:**
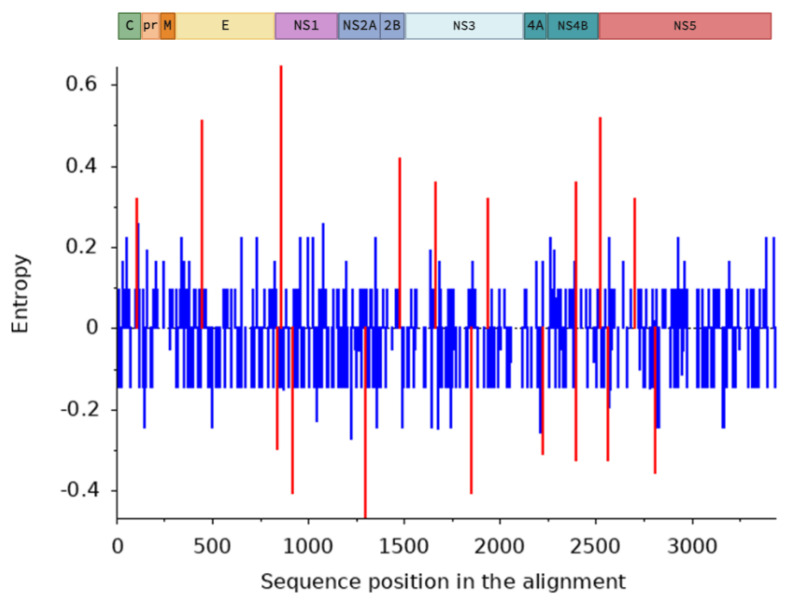
Entropy differences between the “Background” and “Query” sets mapped onto a representation of the WNV genome and its corresponding proteins. Entropy scores > 0 correspond to variable positions that show significant differences between the two sets of sequences. Entropy scores < 0 are relatively conserved or may indicate a lack of divergence between the two sets of sequences. Positions with significant (*p*-value ≤ 0.05) entropy scores are highlighted in red.

**Figure 4 microorganisms-13-00585-f004:**
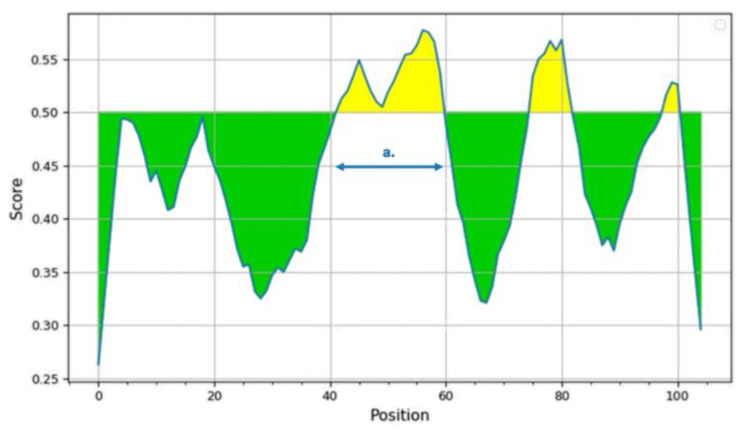
Continuous antibody epitope prediction results. The *Y*-axis shows the residue scores, while the *X*-axis corresponds to the residue positions in the sequence. The (a.) site refers to the predicted antigenic peptide relevant to this study (RGLPIRYQTSAVPREHNGN). Areas in green depict genomic regions with characteristics associated with a low epitope likelihood. Areas in yellow depict genomic regions with characteristics associated with a high epitope likelihood.

**Table 1 microorganisms-13-00585-t001:** Summary of entropy differences and the corresponding amino acid positions (significant sites with a *p*-value ≤ 0.05).

Position	Query Consensus	Entropy Difference Between Background and Query
108	K	0.317
449	I	0.511
838	Q	−0.298
861	S	0.644
919	L	−0.405
1301	L	−0.468
1477	A	0.370
1481	I	0.418
1663	N	0.358
1849	A	−0.405
1941	T	0.317
2224	P	−0.309
2396	A	0.358
2397	Y	−0.325
2522	D	0.516
2561	I	−0.325
2705	R	0.317
2808	K	−0.356

## Data Availability

The data supporting the results of this study can be obtained by contacting the authors; a nucleotide sequence has been submitted to GenBank under the accession number PQ660506.

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
