# Peer review of "Tracking the Pathways of West Nile Virus: Phylogenetic and Phylogeographic Analysis of a 2024 Isolate from Portugal"

_microorganisms, 2025, doi:10.3390/microorganisms13030585_

Round 1

Reviewer 1 Report

Comments and Suggestions for Authors

Dear Authors,

I thoroughly enjoyed reading your manuscript. It is well-written and clearly presents the research in a structured and comprehensible manner. I have only a few minor edits, comments, and suggestions I provided. See below.

Overall, you have done an excellent job in addressing and describing the case in Portugal and the widespread distribution of WNV and the complex ecology associated with this zoonotic disease. I believe this research will be highly valuable for the region and will guide further studies incorporating avian host migration patterns, as well as phylogenetic and phylogeographic analysis.

Best regards,

Line 12: Extra 3

Line 130: Was a negative control included during the extraction?

Line 134: I suggest listing the primer and probe sequences, especially for the RT-qPCR, since there are multiple probes listed in the reference.

Line 140: Barros, S. C. et al., 2013. It is missing in the references.

Line 282: Can the authors provide some information about the CPE, at what day post inoculation was it observed, and the type of CPE? Any photos?

Line 291: Figure 1. I suggest adding the A and B to the figure. It is obvious but needed.  

Line 300: This paragraph is a little vague. Were the sequences 100% identical? I am unsure what the author means by (consistent with each other).

Line 303: Make sure it is available before the article is published.

Author Response

RESPONSE TO EDITOR

We correct an author affiliation (Leonor Orge) and the email address from an author (fabio.abade@iniav.pt instead of fabio.santos@iniav.pt). Additionally, we were able to improve the resolution of Figure 2.

All corrections are highlighted in yellow.

RESPONSE TO THE REVIEWERS:

We express our sincere gratitude to both reviewers for their valuable suggestions and corrections to our manuscript, as well as for the time and effort they devoted to revising our article.

Response to reviewer # 1

Thank you for your kind words and valuable comments and suggestions. We appreciate your insight into improving the manuscript.

Responses:

  • Line 12: Corrected.
  • Line 130: A sentence was added to the Materials and Methods section (2.3): “Simultaneously, as a negative control for the extraction process, a negative sample was subjected to the same treatment as the experimental samples, without any biological material”.
  • Line 134: Suggestion accepted. We added the sequence of the primers and probe in Materials and Methods section (2.3).
  • Line140: Thank you for this correction. Reference added.
  • Line 282: A sentence regarding the CPE was added in the text: “The cytopathic effects (CPE), i.e. cell rounding and ballooning followed by detachment, were identified 3 days after the 2nd passage. Unfortunately, we do not have photos of the VERO cells infected with WNV at the time of the virus isolation.
  • Line 291: You are correct. The suggestion has been accepted, and elements A and B have been incorporated into Figure 1.
  • Line 300: As suggested, we improved the paragraph and rewrote as follows: “The results obtained from both MinION and Illumina sequencing methods were consistent, yielding nucleotide sequences that were 100% identical”.
  • Line 303: Thank you for the reminder. We have already sent an email to NCBI requesting that they release the sequence as soon as possible.

Reviewer 2 Report

Comments and Suggestions for Authors

The manuscript “Tracking the Pathways of West Nile virus: Phylogenetic and Phylogeographic Analysis of a 2024 Isolate from Portugal” by Diogo Maroco and coauthors performed a phylogenetic and phylogeographic study using a WNV strain isolated from a wild bird. Despite being a single case, the study offers valuable genetic information. Overall, the manuscript is well written, the methods have been carefully detailed and well applied. Given the importance of the pathogen, I consider that the manuscript has value for publication. Additionally, I highlight some specific observations.

Introduction
•    Lines 74-81: Indicate, if available, the lineage associated with those reports.
•    Lines 94-106: I suggest restructuring the objectives of the study. Avoid providing results (lineage typing) and discussions (i.e. lines 104-106).
Methods
•    Authors must indicate approval by an ethics committee, or alternatively, the Portuguese National 116 Institute of Agricultural and Veterinary Research (INIAV I.P.).
•    Line 129: Indicate the centrifugation temperature.
•    Line 140: This reference could not be identified. Appropriately indicate the reference in the text as well as in the references section according to the journal guide. 
Results
•    Figure 1: Include definitions (a) and (b) within the figures.
•    Line 280: Indicate the intended reference of the RT-qPCR.
•    Figure 3: The legend should be more detailed. Include the meaning of the lines by color and the chosen thresholds. 
Discussion:
•    The authors must literally state all the limitations of the study (i.e. a unique case, lack of more strains from nearby years from the same country).

Author Response

RESPONSE TO EDITOR

We correct an author affiliation (Leonor Orge) and the email address from an author (fabio.abade@iniav.pt instead of fabio.santos@iniav.pt). Additionally, we were able to improve the resolution of Figure 2.

All corrections are highlighted in yellow.

RESPONSE TO THE REVIEWERS:

We express our sincere gratitude to both reviewers for their valuable suggestions and corrections to our manuscript, as well as for the time and effort they devoted to revising our article.

Response to reviewer #2

Thank you for your valuable comments and suggestions. We appreciate your insight into improving the manuscript.

  • Lines 74-81: As suggested we added information regarding the lineage of WNV detected in pools of Culex pipiens and Cx. univittatus For serological studies, no information is available regarding the virus lineage, as they do not correspond to distinct serotypes.
  • Lines 94-106: We fully understand the reviewer's point of view regarding not providing results in the introduction section. However, if it is not stated in the introduction that the sequenced virus belongs to lineage 1, it is not possible to justify why we conducted the analysis and dataset exclusively on viruses belonging to lineage 1. Therefore, we decided to keep the original text.
  • Ethics approval: Thank you for the reminder. We chose to insert the following sentence in the Institutional Review Board Statement: “The study only used an animal who died after natural infection. No animals were killed or manipulated alive for sample collection for scientific purposes. After death, the animal was sent as part of the routine of the National Reference Laboratory for Animal Health (INIAV, I.P.) and laboratorial investigations were carried out under BSL-3 conditions”.
  • Line 129: Done.
  • Line 140: Corrected.
  • Figure 1: The suggestion has been accepted, and elements A and B have been incorporated into Figure 1.
  • Line 280: Corrected (reference 22).
  • Figure 3: As suggested we improved the legend of figure 3 and we rewrote as follows: “Entropy differences between the “Background” and “Query” sets mapped onto a representation of the WNV genome and its corresponding proteins. Entropy scores > 0 correspond to variable positions that show significant differences between the two sets of sequences. Entropy scores < 0 are relatively conserved or may indicate a lack of divergence between the two sets of sequences. Positions with significant (p-value <= 0.05) entropy scores are highlighted in red.
  • Limitations of the study. We introduced a new sentence at the end of the discussion section: “Furthermore, given that this study is based on a single isolate, this constitutes a limitation, as this sample may not accurately represent the diversity of strains present in circulation within Portugal.